# EnRDeA U-Net Deep Learning of Semantic Segmentation on Intricate Noise Roads

**DOI:** 10.3390/e25071085

**Published:** 2023-07-19

**Authors:** Xiaodong Yu, Ta-Wen Kuan, Shih-Pang Tseng, Ying Chen, Shuo Chen, Jhing-Fa Wang, Yuhang Gu, Tuoli Chen

**Affiliations:** 1School of Information Science and Technology, Sanda University, No. 2727 Jinhai Road, Shanghai Pudong District, Shanghai 201209, China; xdyu@sandau.edu.cn (X.Y.); tsengshihpang@czcit.edu.cn (S.-P.T.); ychen@sandau.edu.cn (Y.C.); wangjf@mail.ncku.edu.tw (J.-F.W.);; 2School of Software and Big Data, Changzhou College of Information Technology, Changzhou 213164, China; 3Jiangsu Zero-Carbon Energy-Saving and Environmental Protection Technology, Yangzhou 225000, China

**Keywords:** semantic segmentation, machine vision, U-Net deep learning, road segmentation, self-driving sweeping bot, residual U-Net

## Abstract

Road segmentation is beneficial to build a vision-controllable mission-oriented self-driving bot, e.g., the Self-Driving Sweeping Bot, or SDSB, for working in restricted areas. Using road segmentation, the bot itself and physical facilities may be protected and the sweeping efficiency of the SDSB promoted. However, roads in the real world are generally exposed to intricate noise conditions as a result of changing weather and climate effects; these include sunshine spots, shadowing caused by trees or physical facilities, traffic obstacles and signs, and cracks or sealing signs resulting from long-term road usage, as well as different types of road materials, such as cement or asphalt; all of these factors greatly influence the effectiveness of road segmentation. In this work, we investigate the extension of Primordial U-Net by the proposed EnRDeA U-Net, which uses an input channel applying a Residual U-Net block as an encoder and an attention gate in the output channel as a decoder, to validate a dataset of intricate road noises. In addition, we carry out a detailed analysis of the nets’ features and segmentation performance to validate the intricate noises dataset on three U-Net extensions, i.e., the Primordial U-Net, Residual U-Net, and EnRDeA U-Net. Finally, the nets’ structures, parameters, training losses, performance indexes, etc., are presented and discussed in the experimental results.

## 1. Introduction

Generally, the behavior of self-driving transportation bots involves smart movements between the warehouse and consumer on a pre-planned GPS route, using sensors for obstacle avoidance. Such an activity is like that of an ant avoiding obstacles using a pre-laid pheromone path when shuttling between its nest and a food source [1]. Analogically, a mission-oriented self-driving bot, such as the Self-Driving Sweeping Bot (SDSB) [2,3], intelligently navigates when sweeping rubbish by means of machine vision which seeks out sweeping targets along a pre-planned GPS route. This may be compared with the scavenger behavior of an herbivorous animal when it seeks out and consumes dead plants for feeding [4]. However, for an SDSB to fully achieve scavenger feeding behavior, many challenges must still be met in terms of machine vision for target detection [5,6,7] and road segmentation outdoors [8,9]. In a modern city, road conditions present a much more complex environment; this includes the presence of many interruptive noises, and the additional factors of changing weather and climate, as well as different kinds of road materials [8].

This work investigated road segmentation for SDSB in terms of a Residual Attention U-Net deep learning approach, in which the input channel applied a Residual U-Net block treated as an encoder, and an attention gate in the output channel was regarded as the decoder; namely, EnRDeA U-Net. The road surface surroundings were considered with respect to intricate noises including differing weather conditions (sunny, rainy, or cloudy), sunshine spots and shadowing caused by large trees or physical facilities, traffic signs, obstacles, and types of road materials (cement or asphalt). The previous work investigated the morphological filtering on the road segmentation [9,10,11,12], in which [9] an optimizing HSV encoding framework is proposed by embedding the morphology operation to explore the road segmentaton. However, during the morphological filtering on segmentation, the different times of corrosion and expansion operatons were iteratively tuned subjectively to reach the optimized results, which implied that the different road optimized segmentation would rely on specific times of corrosion and expansion operatons through user inspection; this fact further indicated that to dynamically tune the parameters on corrosion and expansion operations for road segmentation is infeasible, in terms of the real-time self-driving applications.

To validate the EnRDeA U-Net performance, we first discuss the Primordial U-Net [13] and the Residual U-Net [14]. We then comprehensively examine any advances attained by the EnRDeA U-Net, and present validation as appropriate. The Primordial U-Net mostly adopted the convolutional network architecture used for segmentation of medical and satellite images in a fast and precise manner. In U-Net++ [15,16,17], nested and dense skip connections were introduced to decrease the semantic gap between the encoder and decoder. Three-dimensional U-Net [18,19,20,21] used volumetric segmentation for application to 3D medical scans. Deep Residual U-Net was initially proposed by Zhang et al. [22] to inspect the rich skip connections within networks for information propagation, and this has achieved improved performance. Pavement crack segmentation was investigated by the RUC-net [23] for automatic crack detection. The dResU-Net was applied to brain tumor segmentation from MRI [24]. The dual-attention residual U-Nets integrated two nets for the purpose of transformer oil leakage detection. The Residual Dense U-Net (RDUN) [25] achieved real-world road detection using satellite images. Sea land, rooftops, and iron ore segmentation by the Residual U-Net for remote sensing images was explored for road extraction [26,27,28,29]. Finally, an attention-gated U-Net was successfully validated for lesion segmentation in medical images and remote sensing images [30,31,32].

The Self-Driving Sweeping Bot (SDSB) was investigated in a previous work [2], in which SDSB achieved an intelligent manner of operation, with respect to the security of personnel and materials, as well as sweeping efficiency. For SDSB to achieve self-intelligent movement within a restricted area, various factors for road segmentation must be comprehensively inspected, particularly those relating to varying weather and road conditions. For example, roads are exposed daily to changing weather conditions; these include full sunshine, full cloud cover, and partial cloud cover with sunshine spots and shadowing caused by large trees or physical facilities, as well as different directions of sunshine. This leads to various contrasts in grayscale presentation at the image pixel level for any given region of road. Complex intricate noises must also be taken into consideration. These include cracks and sealing signs as a result of long-term road usage, as well as traffic signs, manhole covers, and roadway speed bumps, all of which greatly affect segmentation efficiency.

In this work, to achieve an SDSB with vision-controllability for operation on restricted roads using a built-in machine vision to segment the sweeping scope in a real-time manner and thus promote sweeping efficiency and security, we validated three U-Net extensions: the Primordial U-Net; the Residual U-Net; and the EnRDeA U-Net. The latter is extended from the Primordial U-Net in terms of its encoder channel, while the Residual U-Net replaces the down-sampling operation of the Primordial U-Net in terms of convolution and pooling layers, so as to increase feature extraction efficiency in a more compact manner. For the decoder channel, the attention block was embedded into the output channel of the Residual U-Net, to assign different weights to each corresponding part of the input feature map, thereby extracting more critical and important information to increase prediction precision. To carry out a detailed analysis of the nets’ features and segmentation performance, and to validate the intricate noises dataset on the three U-Net versions, the nets’ structures, parameters, training losses, performance indexes, etc., are presented and discussed in the experimental results.

The remaining parts of this paper are organized as follows. Section 2 describes the methodology in terms of the EnRDeA U-Net. Section 3 explains the dataset characteristics. Section 4 presents experimental results and analysis. Conclusions are offered in Section 5.

## 2. EnRDeA U-Net Framework

Figure 1 shows the training flowchart of the Residual U-Net embedded with the attention block, in which four layers of the residual blocks (black box) followed up the convolution, batch normalization, and ReLU operations in the input channel, and were then sequentially transposed into the corresponding attention blocks (rosewood color) within the DubleCnv and up-sampling operations for output segmentation. Two extensions of U-nets, i.e., the Residual U-Net and EnRDeA U-Net, were used. In the following sections, their parameter sizes and feature structures, as well as the encoder/decoder flowchart of EnRDeA U-Net methodologies and the principles of residual connection and threshold settings, are also discussed.

### 2.1. Embedded Resdual U-Net with Attention Module

As mentioned above, the EnRDeA U-Net embedded the attention block into the Residual U-Net, in which an encoder located in the input channel and a decoder located in the output channel sequentially performed the learning phases. An encoder can be operated by multiple sequential layers known as the VGG family; for example, VGG11 contains seven convolutional layers followed by the ReLU function [33]. Recently, VGG16 has also been evaluated for pre-training from ResNet-related works [34,35,36]. For the decoder, the transposed convolution layers are constructed so that many sizes of the feature map are yielded and the number of channels relied upon can be reduced in cases such as the LeCun uniform initializer [33]. In this work, the encoder was mainly used for feature extraction, and the decoder used to restore the size of the feature map, with respect to the original image size, by transposing from the residual block into the attention block, to complete the fusion of high-level and low-level features for road segmentation. Details of this process are as follows.

### 2.2. Encoder and Decoder

To validate the encoder channel to increase feature extraction efficiency on road segmentation, residual layers are beneficial to fix the overfitting problem and prevent the degradation of the model. During the training process, as layers increase and deepen, the model is more prone to overfitting, and the loss also tends gradually towards saturation. At the same time, if the network layers continuously increase, so the performance of the model declines. Consequently, the residual connection establishes a residual mapping between input and output during convolution operations, so that the output features contain the encoder features, in terms of the original feature information extracted from residual blocks simultaneously, ensuring that more feature information is retained from the low-level information and transposed into the high-level information, so as to prevent any further gradient loss. In this work, resnet34 was applied and structured as a 34-layer convolutional net for feature extraction on account of its beneficial effect on the U-Net structure. The improvement in encoder and decoder can now be described as follows.

#### 2.2.1. Encoder

The encoder utilized the residual block by replacing the down-sampling operation in terms of convolution and pooling layers from the Primordial U-Net, to increase feature extraction efficiency further, and learn the net in a more compact manner, through the identical mapping of residual connections into the attention block, and thereby alleviate the vanishing gradient problem in backpropagation. Figure 2 shows a typical schematic diagram of the original Primordial U-Net. Although the complexity in the residual block is increased in the network, no more additional parameters are observed, and efficiency is improved. Figure 3 shows two schematic residual blocks. The portion to the left of the dotted line indicates learning of the residual mapping *g*(*x*) = *f*(*x*) − *x*, which makes the identity mapping *f*(*x*) = *x* easier to learn; the portion on the right indicates the input transformed into the desired shape for the addition operation by 1 × 1 convolution.

#### 2.2.2. Decoder

The decoder channel was, thus, embedded with the attention block into the output channel in the Residual U-Net, to extract more representative features in the training phase and thereby increase the segmentation precision. The Residual U-Net retained both low-level feature information and high-level semantic information, and simultaneously connected to corresponding layers between the low- and high-level blocks, to achieve better segmentation results. Sequentially, the Residual U-Net was embedded with the attention block by the addition of the low-level input, with regard to the output, to acquire the weight; this was then multiplied by the low-level feature information and combined with the high-level semantic information. Such an operation is beneficial for assigning different weights to each corresponding part of the input feature map, by enabling the extraction of more critical and important information, and thereby increasing prediction precision. Figure 4 presents a schematic diagram of the attention block embedded in the Residual U-Net.

Inspired by human attention theories concerning the connection between perception and memory [37], which state that the degree of attention paid to an event correlated highly with future recollection of that event, nets have traditionally used convolution operations for feature extraction, maintaining equivalent degrees of attention on each pixel. By embedding the attention block on the net, highly related features regarding the target are retained so that segmentation efficiency can be increased, and irrelevant information can be discarded.

In Figure 4, it can be seen that the weight is firstly multiplied by the deeper layer of the feature map; then, the output is sequentially multiplied by the weight of the shallow feature map, so as to improve the detection of the target area on the weight of the pixel. Features are then combined through the skip layer connection. Figure 4 also shows the detailed operation of the attention module, in which the output $\hat{x}$*^l^_i,__c_* is the multiplied output of $\hat{x}$*^l^_i,c_* with the corresponding attention coefficient *θ^l^_i_*; that is, the attention block embedded in the Primordial U-Net output channel $\hat{x}$***^l^_i,c =_ X^l^_i,c_
***• ***θ^l^_i_***, where *θ^l^_i_* ∈ [0, 1]. This is learned from the parameters automatically to adapt the activation, to precisely segment the road from the surrounding noises and the non-road portions; *l* is the net layer corresponding to each feature, *i* is the pixel space, and *c* is the size of the channel. Calculations of the attention block and coefficients are carried out using Equations (1) and (2)
(1)$$\omega_{attention}^{l}=\varphi^{T}[\sigma_{1}(W_{x}^{T}x_{i}^{l}+W_{g}^{T}g_{i}+b_{g})]+b_{\tau }$$
(2)$$\theta_{i}^{l}=\sigma_{2}(\omega_{attention}^{l})$$

By combining Figure 4 with Equations (1) and (2), it can be determined that the attention block is working by combining the convolution operation of the input feature *x^l^_i_* and the gating signal *g_i_* with the corresponding weights *W_x_^T^* and *W_g_^T^* for the output of $W_{x}^{T}x_{i}^{l}$ and $W_{g}^{T}g_{i}$, respectively. These are then added together via the ReLU to generate $\sigma_{1}(W_{x}^{T}x_{i}^{l}+W_{g}^{T}g_{i}+b_{g})$, which is conducted by the 1 × 1 × 1 convolution to yield $\omega_{attention}^{l}$, to which the Sigmoid *σ*_2_ is then applied to give the final coefficient $\theta_{i}^{l}$. The final feature $\hat{x}$***^l^_i,c_*** is then obtained by multiplying by ***X^l^_i,c_*** and ***θ^l^_i_*** throughout the operation.

## 3. Dataset Analysis

### 3.1. Intricate Road Characteristics

Prior to conducting semantic segmentation by the EnRDeA U-Net on the target road, it was necessary to, first, analyze the characteristics of the collected visual dataset, so that the challenges during segmentation could be understood. Our observation indicated that the roadside area contained large trees, grassy areas, facilities, corridors, and sport parks. The road itself contained several target and noisy objects, including pedestrians, vehicles, bridges, speed bumps, marble blocking balls, traffic signs, and manhole covers. Road and non-road areas were typically separated by retaining walls of 15 cm height. In addition, some road areas were partially or fully overshadowed by large trees and facilities, and many areas contained squiggly black lines, road cracks, and manhole covers. Different types of road material, i.e., cement or asphalt, were also observed, as shown in Figure 5. These factors, combined with changing weather conditions (sunshine, rain, or cloud), meant that road performance was a highly complex presentation. Depending upon location and time, the road could be clouded over, or fully or partially sunlit, and/or overshadowed by trees, facilities, park fencing, etc.

### 3.2. Three Categories of Validated Datasets

In this work, we categorized the collected dataset into three types: sunshine spot shadowed by facilities; sunshine spot shadowed by large trees; and fully cloudy conditions. Examples of these are shown in Figure 6, Figure 7 and Figure 8, respectively. Observations at the pixel level were then analyzed as follows:In fully cloudy conditions, the pixels at the edge between road and grass presented lower grayscale values than in full-sunshine conditions, resulting in confusion with respect to the status of distant road, and failure in segmentation.In full-sunshine conditions, road pixels, on average, exhibited a higher grayscale contrast, compared with non-road areas, particularly if the road was beneath the large trees where partially shadowing sunlight spots occurred. In such cases, the grayscale values of pixels in sunlit areas were obviously higher than in shadowed regions. Moreover, the texture feature on roads lit by intense sunlight caused the region to become more blurred. Additionally, the distant road showed bright regions of high intensity, implying a challenge for road segmentation.Many complex symbols and signs were found that increased the difficulty of road segmentation. For example, there were many cracks in the road caused by long-term intense sunshine. In addition, traffic signs such as white arrows and lines, black manhole covers, zebra crossings, and roadway speed bumps were also influential factors which led to further challenges.

### 3.3. U-Net Extensions Comparison

To evaluate the training cost and the learning efficiency of the Primordial U-Net, Residual U-Net, and EnRDeA U-Net in terms of parameter sizes, we carried out a summarization and comparison, as shown in Table 1. It can be seen that the EnRDeA U-Net required a significantly greater number of training parameters during the training phase. This was because, unlike the other U-Net versions, the EnRDeA U-Net contains both encoder and decoder phases. It may also be observed that the Attention U-Net was particularly beneficial to noise reduction in terms of the three categorized datasets containing many noises (shadows caused by facilities or trees, traffic signs on road, etc.), thereby promoting segmentation efficiency, so that the Attention U-Net required fewer training cost parameters yet achieved a greater segmentation efficiency.

## 4. Experimental Setup and Results

### 4.1. Training Dataset and Data Enhencement

For training and validation purposes, the datasets used in this work, i.e., the Camvid dataset [38] and the self-collection dataset, were collected and evaluated. We found that Camvid contained 11 categories of images regarding pedestrians, cars, roads, etc., in terms of scenes of streets, cement pavements, and urban roads. From a total of 701 images, 600 were used for training and 101 used for validation, each with a resolution of 960 × 720 pixels. From a total of 259 self-collection images, 109 were used, including four types of campus scenes with two kinds of road materials, i.e., cement and asphalt.

Figure 9 presents examples of training images. The Camvid dataset is shown in the first three two-column images in Figure 9a1–c1, and the self-collection dataset in the final three two-column images in Figure 9d1–f1. Within Figure 9, raw images are shown in the first row (Figure 9a1–f1), and corresponding ground-true images in the second row in Figure 9a2–f2. Several backgrounds of scenes surrounding the collected dataset were found. For example, Figure 9a1 shows an urban road; Figure 9b1,d1,e1,f1 show types of campus roads; and Figure 9b1 is a kind of street scene. Many noises can also be observed on roads that significantly influence the semantic segmentation. For example, vehicles running on the road can be seen in Figure 9a1,b1,d1. Speed bumps and white traffic lines can also be observed in Figure 9c1,f1. Several images were inspected showing roads covered by sunshine-shadowed spots, which resulted in an inconsistent pixel-contrast presentation. Different road materials, i.e., the cement road in Figure 9f1 and the asphalt road in Figure 9a1–f1, are also examined in the dataset.

To increase segmentation efficiency in the case of the small-size dataset, one possibility was to increase size depth by convolution kernel to retain more representative features in the dataset; however, such a procedure would have increased the training cost, due to the increasing number of parameters. For this reason, we chose to use data enhancement to improve the diversity of the dataset in a lower cost manner, so that a more robust network could be learned. Specifically, the albumentations toolkit was applied to enhance the dataset in terms of invariant operations including horizontal flipping transformation, translation, and scaling rotation. By such means, the diversity of the dataset was enriched.

In this work, with regard to the loss function, we considered semantic segmentation as a kind of binary classification problem; for this reason, binary cross-entropy was used as the loss function to train the network for road segmentation. Calculation of binary cross-entropy was achieved using Equation (3):(3)$$loss=-\sum \left[ylog\hat{y}\right]+(1-y)log(1-\hat{y})$$
where $\hat{y}$ is the probability of a pixel being classified as a road portion, and *y* is the labelled value of the pixel. The operation is to call the nn. BCEloss() class to calculate the loss in the dataset activated by the sigmoid, and update it through the optimizer by the PyTorch library.

In this work, the Adam optimizer in PyTorch 1.10.0 was applied to train the network. The Adam optimizer offers benefits in terms of its high computation efficiency, low memory usage, and adaptive learning features; these are useful for handling large-scale data and for optimizing parameters. Initially, the learning rate was set to 1 × 10^−4^ and the learning rate was dynamically tuned in terms of exponential decay, the Epoch was set to 120, and batch_size was set to 8. The observation indicated that when the Epoch reached a value of 20, the loss then tended to saturate and terminate to prevent the overfitting condition, so that generalization was increased further. Figure 10 shows the visualized map in terms of the Epoch and training loss. Our observations indicated that when the Epoch reached a value of 100, the loss attained a level of 0.04 and tended towards stabilization.

### 4.2. Segmentation Results and Analysis

To validate the road segmentation efficiency of the three versions of the U-Net extensions (Primordial U-Net, Residual U-Net, and EnRDeA U-Net) in terms of two types of road materials (cement and asphalt) under various conditions (different weather and sunshine/shadowing effects), we used the experimental results shown in the subfigures of Figure 11 as follows: (a) shows a cement road with squiggly black lines and a manhole cover on its surface under fully cloudy weather; (b) shows a cement road surface under sunny conditions with sunshine-shadow spots caused by trees and vehicles beside; (c) shows an asphalt road with speed bumps surrounded and covered on two sides by sparse large trees in bright conditions; (d) shows an asphalt road with speed bumps surrounded by sparse giant trees in bright conditions.

In Figure 11, it can be seen that interference and obstruction in road surface segmentation is caused mostly by pedestrians, vehicles, and fallen leaves, as well as by sunshine spots on the road surface caused by large trees or facilities. Additionally, due to long-term road usage, some cracks, such as the squiggly black lines, were frequently misclassified as traffic markings. Furthermore, the road was generally covered by large numbers of fallen leaves. Such irregular characteristics made it difficult to label the road region, and segmentation was thus impacted by sparse noises; however, this could be alleviated by the morphological operation at a later stage. In intense sunlight, shadowing by the large tree generally caused the road to be covered with many randomly distributed spots, on which the higher gradient of pixel adjacent between the road and the fallen leaves would cause the inconsistency, and representative texture features were lost, and even higher-resolution images could only retain a few, resulting in greater difficulty for road segmentation.

In terms of our detailed analysis of the effects of road noises on segmentation efficiency, we found that the Primordial U-Net and the Residual U-Net performed more poorly in segmenting the road’s distant portion than the EnRDeA U-Net, as can be seen in the red, blue, and green circles, respectively, of Figure 11b. The EnRDeA U-Net increased the weight in the attention gate to suppress the complex distant non-road noises, allowing more attention to be paid to distant-road feature extraction. In the case of an asphalt road in a darker situation, as shown in Figure 11c, the sparse and randomly distributed non-segmented spots obviously appeared in all three U-Net methods, for numerous tiny fallen leaves were randomly distributed and occupied only a small portion of the road, so that grayscale consistency within the road surface was lost. Furthermore, the higher gradient between the fallen leaves and the adjacent road pixels caused the non-segmented spots on the road. In the case of an asphalt road in brighter conditions, as shown in Figure 11d, the Primordial U-Net achieved the poorest segmentation on the brighter regions in terms of the speed bumps and the brighter road regions aside the retaining wall, in which the surrounding minimum grayscale pixel values caused the non-segmented portions shown in red circles. However, the Residual U-Net achieved better segmented efficiency compared with the Primordial U-Net, for the residual block retained more target features via connections between high-level and low-level convolutions in the brighter regions, thereby improving the segmentation efficiency. In the case of speed bumps, the attention gate in the EnRDeA U-Net adjusted the weight to retain more features during the deeper convolution operation, increasing the representative feature, and, thus, improving the segmented efficiency.

### 4.3. Index Evaluation

The performance indexes of PA (pixel accuracy), CPA (class pixel accuracy), and MIoU (mean intersection over union) were used for quantitative evaluation of the segmentation efficiency of the three U-Net versions; of these, PA (pixel accuracy) is more objective and referential than visual evaluation [13]. Table 2 shows that, for the EnRDeA U-Net, the PA, CPA, and MIoU indexes were 96.68%, 95.48%, and 91.77%, respectively. The EnRDeA U-Net demonstrated superior segmentation efficiency compared with the Primordial U-Net in terms of PA, CPA, and MioU, with improved performances of 1.23%, 0.92%, and 2.21%, respectively. Using the same indices, compared with the Res U-Net, the EnRDeA U-Net gave improved performances of 0.64%, 0.34%, and 1.17%, respectively. A follow-up evaluation proved the superior performance of the EnRDeA U-Net model described in this paper, compared with that of the Primordial U-Net and Res U-Net. Figure 12 shows the three indexes of CPA, Recall, and IOU with correspondence to the Epoch in terms of the three U-nets. It can be seen that the EnRDeA U-Net achieved superior efficiency, compared with the other two nets.

## 5. Conclusions and Future Works

In this study, we investigated the proposed EnRDeA U-Net and validated an intricate noise road dataset in which factors affecting segmentation efficiency were considered. These included changing weather conditions, random sunshine-spot shadowing caused by trees or facilities, traffic obstacles and signs, cracks and sealing signs caused by long-term road usage, and different road materials, i.e., cement or asphalt. To comprehensively understand the objective effects of intricate noise on the experimental results, we carried out a detailed analysis and comparison with the Primordial U-Net and the Residual U-Net in terms of the nets’ structure, training loss, parameters, and performance indexes, i.e., PA, CPA, and MioU; these are also presented and discussed in this paper. In the future work, to reach the SDSB with the 24 h/day of road segmentation capability, the road statuses in the night scenery would be considered and collected for training and validation to comprehensively advance and evaluate the model. In addition, the camera sensor captured the images somehow performed the overexposed or underexposed road statuses by the various sunshine directions, which would also be considered for further pre-processing in the dataset, to decrease the sunshine effects and to increase the model training efficiency.

## Figures and Tables

**Figure 1 entropy-25-01085-f001:**
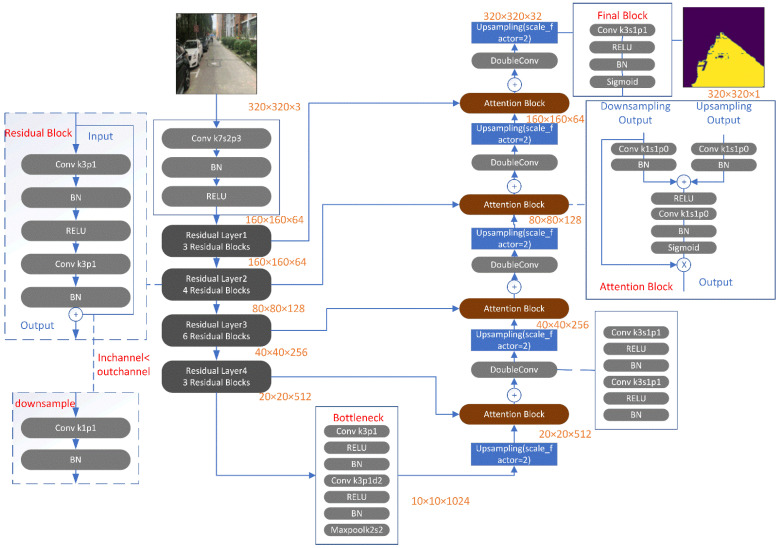
Training flowchart of Residual U-net embedded with attention block, wherein four layers of the residual blocks (black box) for the input original image are sequentially transposed into the corresponding attention gate (rosewood color) for the output segmentation.

**Figure 2 entropy-25-01085-f002:**
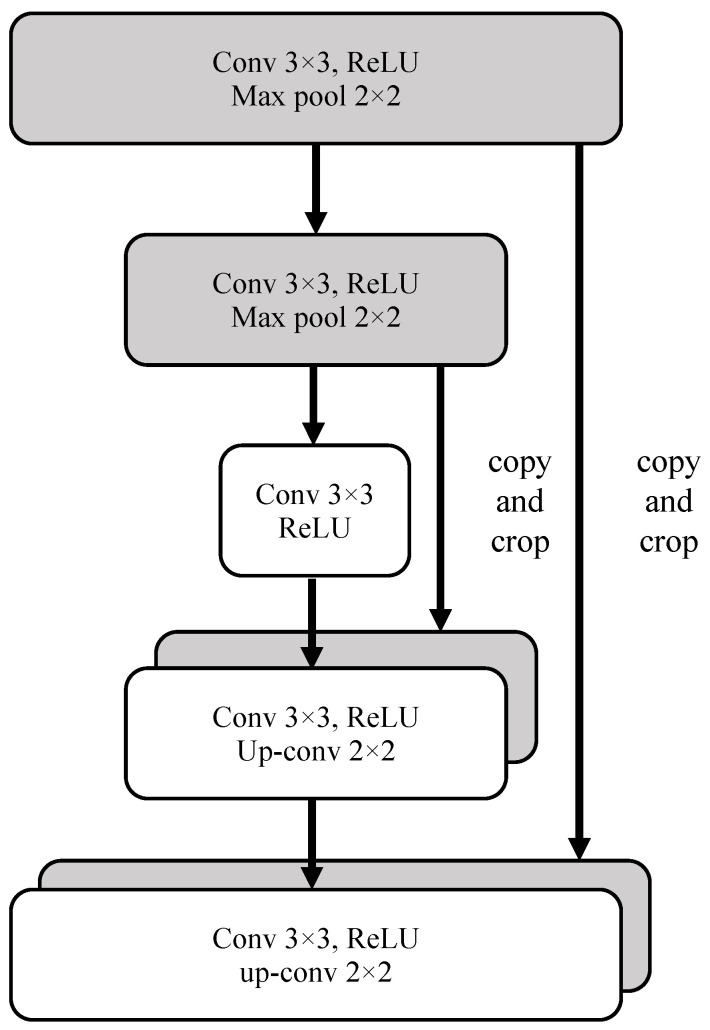
The schematic diagram of primordial U-Net framework.

**Figure 3 entropy-25-01085-f003:**
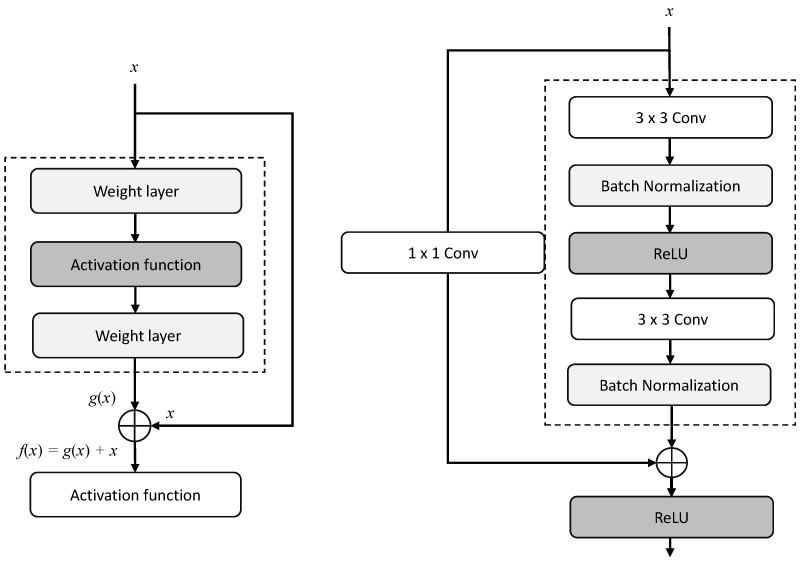
Two schematic residual blocks, the left block of the dotted-line block indicates that the learning of the residual mapping *g*(*x*) = *f*(*x*) − *x*, making the identity mapping *f*(*x*) = *x* easier to learn, and the right diagram indicates the input transformed into the desired shape for the addition operation by 1 × 1 convolution.

**Figure 4 entropy-25-01085-f004:**
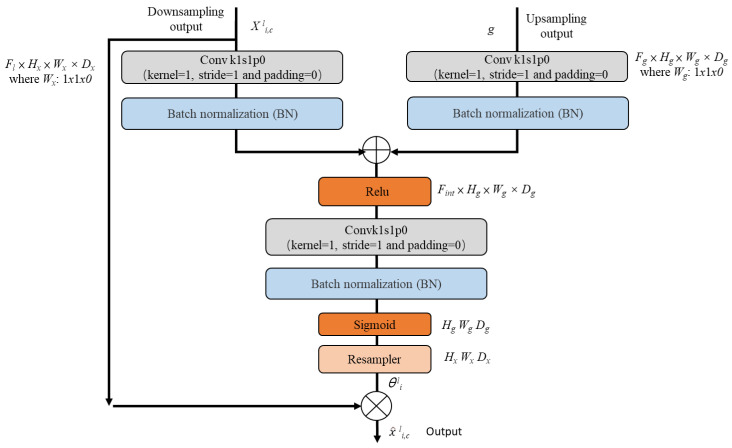
Attention block embedded in primordial U-Net output channel.

**Figure 5 entropy-25-01085-f005:**
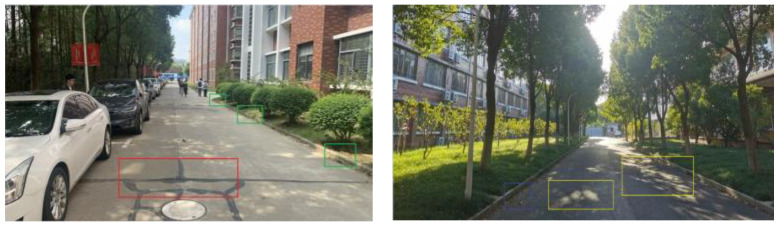
Examples of collected visual dataset, wherein the left image indicated the road with the squiggly black lines (red box) and 15 cm height of the retained wall (green boxes), whereas the right one presented the road surface with the random sunshine spots shadowed by facilities or giant trees (yellow box).

**Figure 6 entropy-25-01085-f006:**
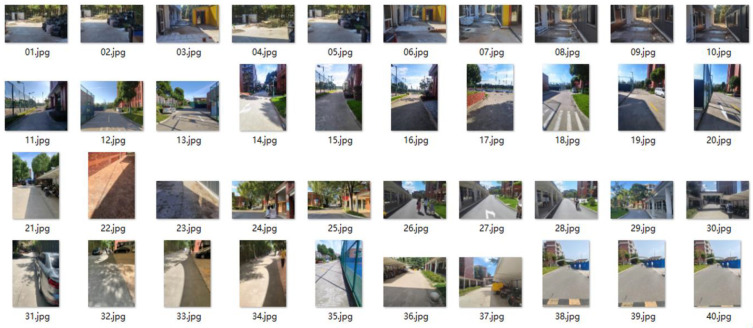
Examples of collected images dataset used for validation, in which the dataset is categorized as road with sunshine spot shadowed by facilities.

**Figure 7 entropy-25-01085-f007:**
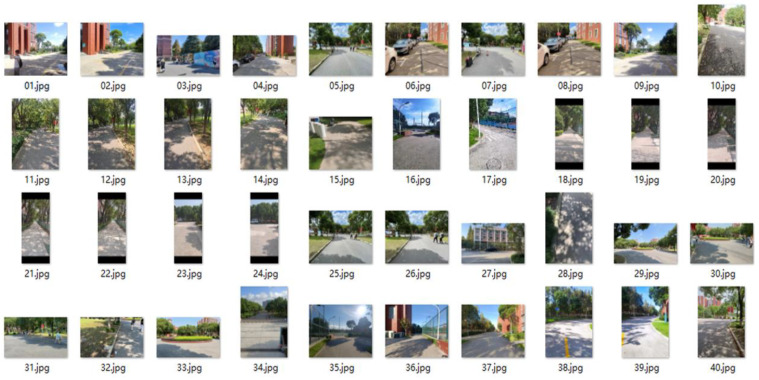
Examples of visual dataset which is categorized as road with sunshine spots shadowed by giant tree.

**Figure 8 entropy-25-01085-f008:**
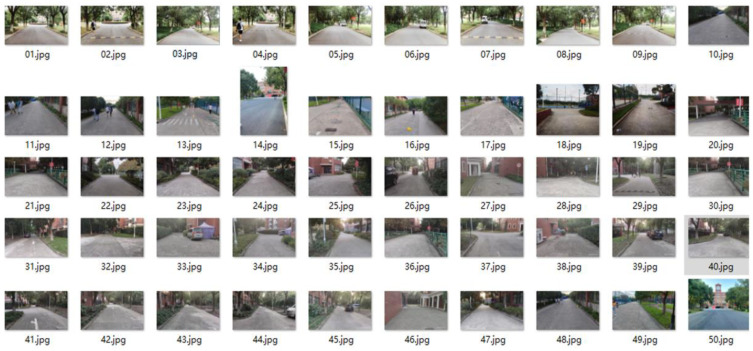
Examples of visual dataset which is categorized as roads under fully cloudy status, wherein two types of road materials, including the cement road and the asphalt road, respectively, are observed.

**Figure 9 entropy-25-01085-f009:**
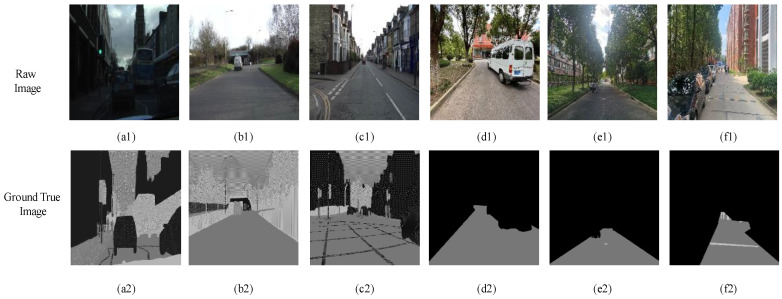
Examples of training images including, Camvid dataset in the first three columns’ images (**a1**–**c1**), and self-collection dataset in the last three columns’ ones (**d1**–**e1**), wherein the first row contains the raw images (**a1**–**f1**), whereas the second row contains the corresponding Ground-True images (**a2**–**f2**).

**Figure 10 entropy-25-01085-f010:**
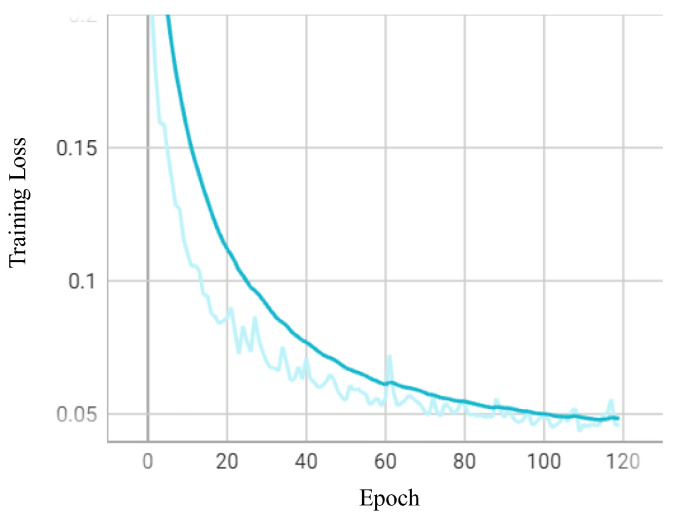
The visualized map of training evaluation in terms of Epoch and training loss.

**Figure 11 entropy-25-01085-f011:**
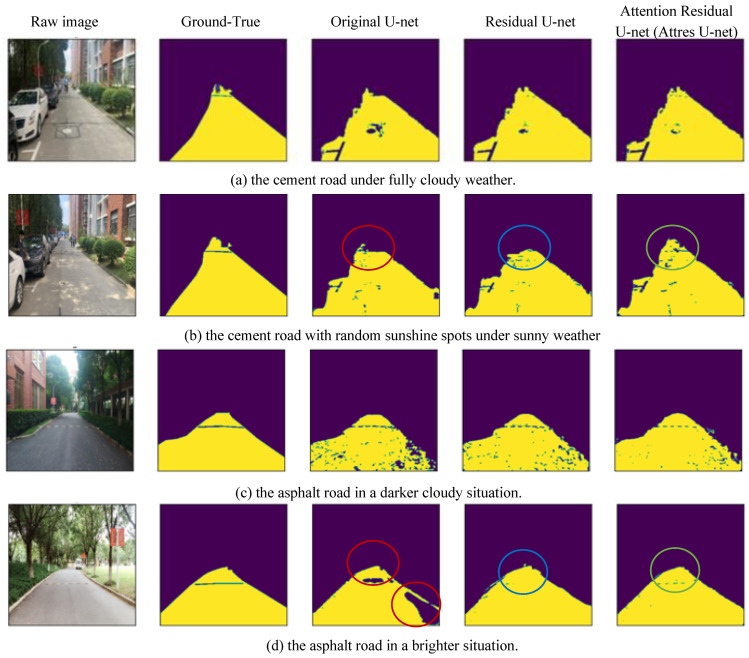
The segmentation experiments on three versions of U-net, including original U-net, Residual U-net, and EnRDeA U-net, validated on four types of road conditions, wherein (**a**) is the cement road under fully cloudy weather, (**b**) is shown as the cement road with sunshine-shadowed spots under sunny weather, (**c**) is indicated as the asphalt road in a darker situation, whereas (**d**) exhibits the asphalt road in a brighter situation.

**Figure 12 entropy-25-01085-f012:**
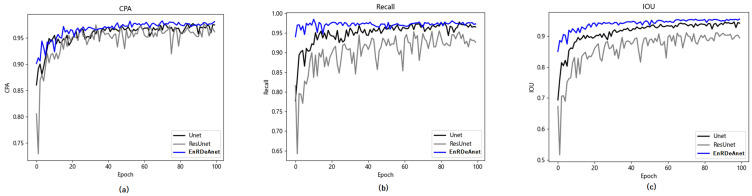
Three indexes of CPA (**a**), Recall (**b**), and IOU (**c**) with correspondence to Epoch number in terms of three U-nets, in which the EnRDeA U-net (blue line) significantly performed the superior efficiency compared to the other two (black line and gray line).

**Table 1 entropy-25-01085-t001:** Summarized Training Performance Comparison of Four Versions of U-net Extensions.

Net Versions	U-Net	Residual U-Net	Attention U-Net	EnRDeA U-Net
Net Structure	Encoder-coder	Encoder with Residual Block	Decoder with Attention Gate	Pairwise Encoder/Decoder with Residual Block and Attention Gate
Number of Parameters	34.53 M	48.53 M	34.88 M	52.02 M
Characteristic Performance	Original Method Lower Training Cost	Better Segmented Efficiency, Higher Training Cost	Noises Reduction, Lower Training Cost	Superior Segmented Efficiency, Fairly Training Cost

**Table 2 entropy-25-01085-t002:** Model Metrics Evaluation.

U-Net Model Name	PA	CPA	MIoU
U-Net	95.45%	94.56%	89.56%
Residual U-Net	96.04%	95.14%	90.60%
EnRDeA U-Net	96.68%	95.48%	91.77%

## Data Availability

Not applicable.

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
