# Peer review of "EnRDeA U-Net Deep Learning of Semantic Segmentation on Intricate Noise Roads"

_entropy, 2023, doi:10.3390/e25071085_

Round 1
Reviewer 1 Report
The manuscript presents an investigation on the proposed EnRDeA U-Net model and an intricate noise road dataset was validated in which factors affecting segmentation efficiency were considered. Please address the following comments and questions:
1. On Line 33, it should be “to and from between warehouse… ”.
2. Is the validation presented in the paper enough to validate the efficiency and accuracy of the model.
3. Following Question 2, what are the other validations that could be done?
4. What are the drawbacks of the proposed EnRDeA U-Net model in the study? Please address this question in detail.
Overall, the quality of English is okay. Some minor editing should be done before the paper is published.
Author Response
Response to Reviewer 1 Comments
Comments and Suggestions for Authors
The manuscript presents an investigation on the proposed EnRDeA U-Net model and an intricate noise road dataset was validated in which factors affecting segmentation efficiency were considered. Please address the following comments and questions:
Point 1: On Line 33, it should be “to and from between warehouse… ”.
Response 1: Thank you for the comment, on Line 33, the sentence has been revised as “to and from between warehouse” in the revised version.
Point 2: Is the validation presented in the paper enough to validate the efficiency and accuracy of the model.
Response 2: Thank you for the insightful comment, for training and validation purposes, the datasets used in this work, i.e., the Camvid dataset [29] and the self-collection dataset, were collected and evaluated. We found that Camvid contained 11 categories of images regarding pedestrians, cars, roads, etc., in terms of scenes of streets, cement pavements, and urban roads. From a total of 701 images, 600 were used for training and 101 used for validation, each with a resolution of 960 * 720 pixels. From a total of 259 self-collection images, 109 were used, including four types of campus scenes with two kinds of road materials, i.e., cement and asphalt. In addition, the validation considered the training cost between the training loss with Epoch number, our observations indicated that when Epoch reached a value of 100, the loss attained a level of 0.04 and tended to stabilization. Above validation can be regarded as a preliminarily step for the model evaluation. To reach the segmentation in a highly efficient manner, routinely and iteratively collecting the road conditions under the complex conditions outdoors are essential to continuously train the update dataset and validation is a practical way for usability in the real-world application. Above explanation has been appended into the revised version on line 366 to 375 at the conclusion and the future works.
Point 3: Following Question 2, what are the other validations that could be done?
Response 3 : Thank you for the insightful comment. In the future work, to reach the SDSB with the 24 hrs/day of road segmentation cabability, the road statuses in the night scenery would be considered and collected for training and validation to comprehensively advanced evaluate the model. In addition, the camera sensor captured the images somehow performed the overexposed or underexposed road statuses by the various sunshine directions, which would be also considered for further pre-processing on dataset to decrease the sunshine effects and to increase the model training efficiency. Above explaimation has been appended into the revised version on line 366 to 375 at the Conclusion and Future Works.
Point 4: What are the drawbacks of the proposed EnRDeA U-Net model in the study? Please address this question in detail.
Response 4: Thank you for the insightful comment, the proposed EnRDeA U-Net model is typically a deep learning framework, in which the road regions in the image are labelled to generate the ground-true image prior to the dataset training and prediction. However, during the labelling step, many obstacles, and noises e.g., pedestrians, vehicles, traffic cones, speed bumpers and marble balls etc., interfered the labelling, such that increases the difficulties for correctly labelling the ground-true road region, further to influence the road segmentation efficiency. In addition, the qualities of captured image on camera regarding the road regions are often affected by the sunshine direction to generate the overexposed or underexposed images, led to hardly and correctly segment the accurate road region. In such a case, the overexposed or underexposed images will be dealt with by means of image processing prior to the learning steps to increase the training-model precision.

Reviewer 2 Report
In traditional computer vision, this type of noise is straightaway removed using morphological filtering. Why is it necessary to use deep learning for every simple noise reduction that can be achieved with simple morphological filtering? With morphological filtering, usually noise reduction result in 100% success where neural network fails to provide such success.
Please justify why morphological filtering cannot be used here to resolve this research problem. It is not worth spending so much resources on a non-existing problem.
The English language used here, even though correct in grammar and sentence structure, can be still written in a more cohesive and a simple flowing way.
Author Response
Response to Reviewer 2 Comments
Comments and Suggestions for Authors
Point 1: In traditional computer vision, this type of noise is straightaway removed using morphological filtering. Why is it necessary to use deep learning for every simple noise reduction that can be achieved with simple morphological filtering? With morphological filtering, usually noise reduction result in 100% success where neural network fails to provide such success.
Response 1: Thank you for the insightful comment. The method of the morphological filtering for road segmentation was investigated in the previous work [33], wherein an optimizing HSV encoding framework is proposed by embedded the morphology operation to explore the road segmentation. However, during the morphological filtering on the road segmentation, the different times of corrosion and expansion operations would be iteratively tunning subjectively to reach the optimized road segmentation in an image, such a fact implied that the different road segmentation would be relied on specific times of the morphological operations through user inspection, the fact indicated that to dynamically tune the the morphological operations for road segmentation would be infeasible in terms of the real-time self-driving application. On the contrary, the proposed EnRDeA U-Net model of deep learning approach for the road segmentation, of which the efficiency is highly relevant to the involved training dataset in relating with the labelled ground-true image. Such facts implied the efficiency of the proposed EnRDeA U-Net model can be reached the generalization perspective. Above explanation has also appended into the revised version on line 52 to 61 at Introduction part with a new reference [34] appended at References part.
Point 2: Please justify why morphological filtering cannot be used here to resolve this research problem. It is not worth spending so much resources on a non-existing problem.
Response 2:
- Thank you for the insightful comment, as aforementioned the response at Point 1, the morphological filtering for the road segmentation implied that the different road segmentation would be highly relied on the specific interative times of corrosion and expansion operations through user inspection, led to infeasibly applied in the real-time self-driving application.
- Currently an SDSB to fully achieve scavenger feeding behavior, many challenges must still be met in terms of machine vision for target detection and road segmentation outdoors. For a modern city, road conditions present a much more complex environment; this includes the presence of many interruptive noises, and the additional factors of changing weather and climate, as well as different kinds of road materials, such the existing problems have to be further overcome to reach the sweeping efficiency and security on SDSB.
- Above explanation has been appended into the revised version on line 52 to 61 at Introduction part with four new references [33-36] appended at References part.
Point 3: The English language used here, even though correct in grammar and sentence structure, can be still written in a more cohesive and a simple flowing way.
Response 3:Thank you for the comment, to increase the readability on the manuscript, the English proofreading of the manuscript has been conducted by MDPI English editing service on May-21, 2023, ID:66498. A more cohesive and a simle flowing way on the manuscript will be considered and followed up by the next new submission.

Round 2
Reviewer 1 Report
Good, all the questions have been answered.
Reviewer 2 Report
The authors have responded to the reviewers comments sufficiently.